# The Role of Chitosan and Graphene Oxide in Bioactive and Antibacterial Properties of Acrylic Bone Cements

**DOI:** 10.3390/biom10121616

**Published:** 2020-11-30

**Authors:** Mayra Eliana Valencia Zapata, Carlos David Grande Tovar, José Herminsul Mina Hernandez

**Affiliations:** 1Grupo de Materiales Compuestos, Escuela de Ingeniería de Materiales, Universidad del Valle, Calle 13 # 100-00, Cali 76001, Colombia; valencia.mayra@correounivalle.edu.co; 2Grupo de Investigación de Fotoquímica y Fotobiología, Universidad del Atlántico, Carrera 30 Número 8-49, Puerto Colombia 081008, Colombia

**Keywords:** acrylic bone cement, antibacterial activity, bioactivity, chitosan, graphene oxide, poly(methyl methacrylate)

## Abstract

Acrylic bone cements (ABC) are widely used in orthopedics for joint fixation, antibiotic release, and bone defect filling, among others. However, most commercially available ABCs exhibit a lack of bioactivity and are susceptible to infection after implantation. These disadvantages generate long-term loosening of the prosthesis, high morbidity, and prolonged and expensive treatments. Due to the great importance of acrylic bone cements in orthopedics, the scientific community has advanced several efforts to develop bioactive ABCs with antibacterial activity through several strategies, including the use of biodegradable materials such as chitosan (CS) and nanostructures such as graphene oxide (GO), with promising results. This paper reviews several studies reporting advantages in bioactivity and antibacterial properties after incorporating CS and GO in bone cements. Detailed information on the possible mechanisms by which these fillers confer bioactive and antibacterial properties to cements, resulting in formulations with great potential for use in orthopedics, are also a focus in the manuscript. To the best of our knowledge, this is the first systematic review that presents the improvement in biological properties with CS and GO addition in cements that we believe will contribute to the biomedical field.

## 1. Introduction

Acrylic bone cements (ABC) are widely used in orthopedics for joint fixation [1,2], antibiotic release [3], bone defect filling, fractures stabilization, screw augmentation, cranioplasty, vertebroplasty, and kyphoplasty [4,5,6]. One strategy used to provide ABCs with bioactivity is the addition of CS, which, thanks to its excellent biological properties, transfers to the ABC the capacity to generate chemical bonding with the bone in the biological environment, achieving a more stable fixation through the improved interlocking between the bone and cement. On the other hand, GO has demonstrated a high degree of mechanical reinforcement and activity against a wide range of bacteria, which generates antibacterial properties in the cements and reduces the risk of septic loosening of the implants. Recently, a review was written about the biological response of cements added with GO [7]. However, the purpose of that review included a wide range of properties (physical, mechanical, and biological), and there was an in-depth analysis of the biological properties.

Due to the great importance of acrylic bone cements in orthopedics, many research methods have improved their bioactivity and antibacterial activity. Figure 1 shows the ascending tendency in the number of publications per year in cements added with GO and the high importance in the last years concerning the study of ABCs added with CS, demonstrating the tremendous present interest in research in these topics.

## 2. Arthroplasty

Osteoarthritis (also known as osteoarthrosis or degenerative joint disease) is the most common form of arthritis. The knee and hip are the joints most affected by this disease. When a joint develops osteoarthritis, the cartilage is damaged, worn out, or broken, and the exposed bones friction causes pain, swelling, loss of joint movement, and deformities [8].

Arthroplasty has become consolidated as a surgical procedure to improve people’s quality of life with osteoarthrosis or joint trauma by improving physical function and reducing pain. Worldwide, the number of operations of this type has increased over the years, and a more significant increase is expected in the next few years by the increase in the aging of the world’s population [9].

In total joint replacement (TJR), also known as arthroplasty, the diseased or damaged parts of the joint are removed (Figure 2a) and replaced by prostheses or implants (Figure 2b). TJR can be performed on various joints, including the hip, knee, ankle, shoulder, elbow, fingers, and wrist. However, hip and knee prostheses are the most common [10].

During TJR, the most common method of fixing the implant is with a load-transfer material’s introduction, typically an acrylic bone cement (ABC) (Figure 2c), introduced into the space between the implant and the joint as a flow-able mixture that eventually hardens over time [11]. The main advantage of these cemented joint replacements is the reduction in the recovery time of the surgery. Once polymerized, the cement can support a load and offer immediate stability, and it has shown excellent long-term results [12]. However, if the cement layer is loosened, the surrounding bone can be reabsorbed, and implant failure can eventually occur. Today, acrylic bone cements use is around 90% of total hip replacement (THR) surgeries in developed countries such as the United Kingdom, the Netherlands, and France.

## 3. Acrylic Bone Cements (ABC)

Most of the commercial ABCs available today consist of two components, one solid, based mainly on poly(methyl methacrylate) (PMMA), and one liquid, based on methyl methacrylate (MMA) [13], which are mixed and, through the polymerization reaction of the monomer, transformed into a hardened cement paste [11]. Figure 3 shows the presentation of Surgical Simplex P commercial cement.

PMMA cements are the most commonly used to bond and load transfer between the implant and the bone [14,15]. The main advantage of using it is the excellent primary fixation obtained between the implant and the bone and the patient’s faster recovery [16]. On the other hand, PMMA is a fragile material with a low resistance to fracture and low fatigue life [11].

Commercial ABCs do not differ drastically between brands. The main differences between them are the addition of PMMA copolymers or antibiotics into the solid phase, comonomers to the liquid phase, solid/liquid ratio variation, radiopaque agent variation, or additives chlorophyll Palacos^®^ [11,17]. The necessary components of the ABCs are shown in Table 1.

The polymerization process begins after mixing of the benzoyl peroxide (BPO) (in the solid) and *N,N*-Dimethyl-*p*-Toluidine (DMPT) (in the liquid) with the production of benzoyl radicals at room temperature [18,19]. The heat produced during polymerization is between 52 and 57 kJ per mole of MMA [3]. The maximum temperature reached during the polymerization reaction is known as T_max_ [20], exceeding 100 °C. These high temperatures can cause cellular bone necrosis and contribute to aseptic loosening [10,21,22,23]. Polymerization temperatures experienced under in vivo conditions are much lower (between 40 and 47 °C) at the bone interface due to the reduced thickness of the bone cement layer used in TJR, the presence of blood circulation, and heat dissipation through the implant and surrounding tissue [19,24,25,26].

The leading cause of failure of cemented arthroplasties is aseptic loosening of the prosthesis, which usually occurs at the bone–cement interface and requires a second surgery to replace the whole system [27]. The loosening occurs according to:High polymerization temperature of the cements (between 67 and 124 °C) [28], which generates thermal necrosis of the bone [29], alteration of the local blood circulation, and predisposition to the formation of a fibrous membrane in the bone—cement interface [16,30].Release of unreacted residual monomer or MMA, which generates chemical necrosis of the bone.Contraction of the cement during polymerization.A significant difference between the cement’s stiffness and the adjacent bone generates an inappropriate load transfer [27].Interaction of the cement particles with the surrounding tissues, which produces the inflammatory responses of the periprosthetic tissue and increased bone destruction.Lack of osseointegration due to its inert nature [14].

All these disadvantages generate a lack of strong cement–bone interaction as the only adhesive force is the interdigitation of the cement with the bone, without any apparent chemical reaction. Therefore, fibrous tissue is encapsulated, causing instability, and movements in the bone—cement—prosthesis interface, which is considered the weak bonding zone. These micromovements can accelerate aseptic loosening, causing implant failure [14,31].

Since a substantial fixation of the ABCs to the bone depends primarily on mechanical anchorage [32], many investigations in two directions are ongoing to overcome loosening in the prosthesis. The first consists of generating bioactivity of the cement, which is a critical factor in achieving long-term stability of the implant [14], since chemical bonding is achieved between the bone and the cement, while the second consists of providing antimicrobial properties to the ABCs since infections are common in arthroplasty and also lead to septic loosening.

## 4. Chitosan (CS)

CS is a linear, semicrystalline polysaccharide and the direct derivative of chitin, the second most abundant natural polymer after cellulose [33]. It is a polyelectrolyte with reactive functional groups [34]. Chemically, CS is a copolymer of β-(1-4) D-glucosamine (deacetylated unit) and *N*-acetyl-*D*-glucosamine (acetylated unit) in different proportions, which depend on the degree of deacetylation (DD) [34,35]. As shown in Figure 4, this polycationic polymer has an amino group and two hydroxyl groups in each glycosidic unit. The positive charge through the polymer chain allows many electrostatic interactions with negatively charged molecules [36]. Structurally, it is similar to glycosaminoglycans, the main component of the bone’s extracellular matrix [37].

The presence of some functionalities such as -NH_2_ and -OH in CS molecules provides the basis for interaction with other polymers and biological molecules [38,39]. The amino group’s functionality gives rise to chemical reactions such as acetylation, quaternization, alkylation through reactions with aldehydes and ketones, copolymerization, and chelation of metals to provide a variety of products with different properties. The two hydroxyl groups in each glycoside unit allow several reactions such as *O*-acetylation, hydrogen bonds with polar atoms, and grafting, among others [40].

At low pH, the protonation of the primary amino group is present in CS, generating a positive charge that converts CS into a cationic polyelectrolyte, causing electrostatic repulsion between the polymer chains, which facilitates its solubility [41].

The physical properties of CS depend on parameters such as molecular weight (Mw) and deacetylation degree (DD) (in the range of 50–95%) [40,41]. It is biodegradable, biocompatible, promotes cell growth and the deposition of a mineral-rich matrix by osteoblasts, is easily soluble in diluted acid solutions with pH less than 6, is antioxidant, presents antimicrobial, antifungal, and antitumor activities, is mucoadhesive, and is analgesic [34,38,39,40,41,42,43,44,45,46].

## 5. Graphene Oxide (GO)

GO is a two-dimensional material that consists of an oxidized form of graphene with hydroxyl, epoxy, and carbonyl groups on its basal plans and carboxylic groups at its edges [47,48,49,50] (Figure 5). Despite the defects that these groups generate in the graphene sheets, the oxidized sheets keep almost intact their mechanical properties showing Young’s modulus values as high as 0.25 TPa. In addition, GO is biocompatible [51,52,53], biodegradable [54,55,56,57], and sensitive to temperature and pH changes for drug release [58], has excellent antibacterial properties [49,59,60,61,62], has a high specific surface area [63], is impermeable to all gases and liquids except water [64], has physiological stability [51], and has easy biological/chemical functionalization [65]. The functional groups of graphene oxide improve the dispersion of graphene, and they also improve its solubility in water and some organic solvents [64,66], possessing photoluminescence in the wavelength range of the near-ultraviolet and the visible blue to the near-infrared [66].

## 6. Bioactivity

Bioactive materials interact with bone tissue when implanted inside the bone to stimulate a biological response from the body, such as bonding to tissue and promoting osteogenesis [67,68]. There are three kinds of bioactive materials: osteoconductive, osteoinductive, and osteogenic. Osteoconductive materials support the attachment of new osteoblasts and osteoprogenitor cells, providing an interconnected structure that stimulates bone growth along with the bioactive material [69,70]. Osteoinductive materials induce nondifferentiated stem cells or osteoprogenitor cells to differentiate into osteoblasts [69] and stimulate the growth of new bone on the material away from the bone/implant interface [70,71]. Osteogenic refers to osteoblasts at the new bone formation producing minerals to calcify the collagen matrix [69].

Bioactive materials have the following functions [69]:Stimulate cell differentiation and proliferationStimulate gene and tissue regenerationRelease bioactive molecules actively and effectively for restoring and repairing the impaired functionality of the organs.

Thanks to these desirable bioactive materials, they have become critical in tissue engineering [69].

### 6.1. Bioactivity in Acrylic Bone Cements

Bone cement is a biologically inert component, so it does not usually promote bone growth [10]. Different strategies look to increase the bone and the cement’s interfacial resistance and improve the prosthesis’s performance. One of these strategies consists in the incorporation of bioactive fillers in the solid phase of the cement to generate ABC with a bioactive surface, capable of promoting direct bone apposition instead of encapsulation of the implant by fibrous tissue, promoting bone growth and the formation of a strong chemical bond between the cement and the bone [14,32].

Some bioactive fillers used are hydroxyapatite (HA) [72,73,74,75], phosphate bioglasses [29,76], silicate and borate bioglasses [30,77], tricalcium phosphates [31,78], and *Sepia Officinalis* cuttlebone [79]. With this type of inorganic filler, it is necessary to control the amount used since they affect the cement’s mechanical properties [32,75]. Besides, most of them tend to make the cement even more brittle, and the matrix must be modified to increase its ductility and decrease its modulus [27].

Some modifications to the matrix have been achieved with the addition of comonomers in the liquid phase, such as dimethyl aminoethyl methacrylate (DMAEM) [74], diethyl amino ethyl acrylate (DEA) [80], acrylic acid (AA) [73], 4-methacryloxyloxybenzoic acid (MBA), methacrylic acid (MAA), 4-dimethylamino benzyl methacrylate (DEABMA) [81], 5-hydroxy-2-methacrylamidobenzoic acid (5-HMA), and 2-hydroxyethyl methacrylate (HEMA) [82]. The modifications adding other polymers to the solid phase such as hydroxypropyl methacrylate (HPMA) [83] and core–shell nanoparticles with polybutyl acrylate (PBA) core and PMMA shell [84] have also been reported, allowing to modify the stiffness of the matrix and increase properties such as impact resistance.

Bone resorption has been reported after prolonged implantation of some bioactive cements, which compromises the prosthesis’s fixation due to the wear of the weak calcium phosphate layer formed on the cement’s surface, generating particles that stimulate bone resorption [32,85].

The presence of biodegradable substances in the ABCs generates partially biodegradable cements, which allow the replacement and growth of new bone inside the cement in the pores produced by the hydrolytic and enzymatic degradation in the biological environment, thus improving the mechanical anchorage [32,77,86] or generating systems of controlled drug release [15].

Biodegradable polymers such as polylactic acid (PLA), β-polyhydroxy butyrate (PHB), and thermoplastic starch (TPS) have been added to the solid phase in a range between 49 and 66 wt.% to create partially biodegradable ABCs, finding in all cases, a decrease in the mechanical properties of the cement [15,87]. There are studies where percentages from 10 to 30 wt.% of beads composed of PMMA/ε-polycaprolactone were added. The addition of 10 wt.% produced increased bending properties, while 30 wt.% decreased properties in bending and compression [88]. Other researchers report that a PHB copolymer and hydroxyvalerate (PHBV) addition increased the mechanical properties compared to PHB and the cement’s biological response [77]. Besides, gelatin microparticles improved the release of antibiotics, leading to decreased mechanical properties [89]. In all cases, these biodegradable polymers’ presence increased the cement’ water absorption [90]. A recent review of partially degradable acrylic cements published information about modifications, advantages, and disadvantages of cement [91]. 

The combination of porosity and bioactive agents in the cement can present a synergistic effect improving cell colonization in the material, osteoblasts activity, osteoinduction processes, and osteoconduction for bone regeneration [92]. The mixture of porosity and bioactive agents is possible, adding CS and a bioglass, but these cements’ mechanical properties are low to be used in orthopedic applications [32].

### 6.2. Bioactivity in Acrylic Bone Cements Loaded with Chitosan

Although CS presents similarities with the extracellular matrix and has demonstrated some intrinsic bioactivity, its use as the only material in bone tissue engineering is limited by an inherent polysaccharide’s low mechanical properties with an eventual loss of biological properties over time [93,94].

Several investigations about CS addition’s effect reported cement with improved mechanical and thermal properties with chitosan additions from 5 to 10 wt.%, depending on the particle size used [4,16,95]. Additionally, the presence of CS at 5 wt.% contributed to avoiding the formation of the fibrous tissue interface between the bone and the implanted cement in bone defects of the animal model (rat), improving the bone–cement integration [16], and generating a significant increase in the osteoblast cellular activity compared to the ABC without chitosan [95]. In contrast, other researchers found decreased mechanical properties with CS’s addition to the ABCs [96,97].

The inclusion of 10 wt.% chitosan in the solid phase of an ABC with different HA-proportions generated pore spaces for osteoconduction. However, losses in the compressive strength of up to 46.3% were presented [92]. Lin et al. [98] reported the use of calcium β-phosphate microspheres encapsulated in chitosan gel. Zamora et al. [99] used CS microspheres in the solid phase of bone cements, which generated lower maximum curing temperatures (T_max_). They increased setting time (t_set_) and also produced a rough surface due to its gradual degradation, which benefited adhesion and cell proliferation but reduced mechanical properties. Besides, Zamora et al. [99] reported subdermal implantation studies in Wistar rats, concluding that the cements had excellent compatibility.

The presence of up to 20 wt.% of CS, besides, favored the bioactivity of the ABCs by increasing the Ca and P ions deposited on its surface when being immersed in a simulated biological fluid (SBF) for different periods [100], decreased the T_max_ and increased the t_set_, and improved the surface wettability [45,100]. Additionally, the implantation in Wistar rats’ parietal bone showed an increase in the cement’s osseointegration with CS. This osseointegration was higher when the formulation presented CS + GO, with which a near-total healing bone–cement interface was reached in the bone defect [101]. However, it was evidenced by the reduction in the mechanical properties and the increase in the residual monomer content.

Tavakoli et al. [102] found that incorporating 25 wt.% of a CS/GO nanocomposite in the solid phase of the cements improved healing properties, increased bioactivity after 4 weeks of immersion in SBF, and increased cell viability, growth, and cell adhesion in MG-63 cell culture. Similar results were obtained by Soleymani et al. [103], when CS/multiwalled carbon nanotubes composite was incorporated in the formulation of ABC.

### 6.3. Bioactivity in Acrylic Bone Cements Loaded with Graphene Oxide

Graphene oxide has attracted attention in bone tissue engineering applications because of their attractive biological, mechanical, and physically properties mentioned before [104,105]. The presence of polar groups on the GO surface improves compatibility with polymer matrices [52,106].

The addition of GO in percentages up to 0.5 wt.% in ABCs has favored bending and compression [107,108,109,110]. Paz et al. [110] reported increased fracture toughness and fatigue performance attributed to the GO, induced deviations in the crack fronts, and hampered crack propagation.

Several authors reported that the increase in GO content in ABCs decreases the maximum polymerization temperature. It acts as an inhibitory and retardant agent of the MMA polymerization reaction [107,108,110], which decreases the risk of thermal necrosis and improves the cement’s bioactivity. On the other hand, this inhibitory effect on the polymerization reaction has generated an increase in the cement’s residual monomer content, which increases the possibility of chemical necrosis of the surrounding tissue with percentages of GO higher than 0.3 wt.% [108].

ABCs loaded with GO presented high cell viability, low apoptosis, and extensive spread on disc surfaces of mouse L929 fibroblasts and human Saos-2 osteoblasts [111]. Pahlevanz et al. [112] reported that bone cement bioactivity consisting of PMMA and polycaprolactone (PCL) was improved with fluorapatite and GO, evidenced by an increase in both in apatite formation ability on the polymer surface and in cell viability against MG-63 osteoblast.

Mirza et al. [113] prepared ABCs with GO and evaluated human bone marrow mesenchymal stem cells (hBMSCs). The anabolic genes (COL1A1, BMP4, BMP2, RUNX2, and ALP) were associated with stimulatory effects, while the catabolic genes (MMP2 and MMP9) exhibited inhibitory effects in the presence of GO. They found early osteogenesis and increased proliferation of hBMSCs on the formulation with higher GO content (0.048 wt.%) compared to the control and lower GO content. On the other hand, Paz et al. [114] evaluated MC3-T3 viability after an incubation period of 72 h when they were exposed to cements with 0.1 wt.% GO and without any cytotoxic effect present.

Valencia et al. [109] reported that GO’s presence did not generate any cytotoxic effect in ABCs. The same group [101] implanted GO-loaded cement in the parietal bone of Wistar rats. They reported that GO improved the osseointegration of the cement since it accelerated the sealing process.

Contrary to all the results shown above, Sharma et al. [22] compared cements’ osseointegration with GO, graphene (G), and amine-functionalized graphene (AG); showing a reduction in cell viability against human osteoblast (MG-63) in the ABCs loading with GO and G compared to the cement containing AG. Then, cell apoptosis was generated because GO and G penetrated the cytoplasm by endocytosis pathway, disturbing the metabolic activity, gene transcription, and translation processes.

The summary of the bioactivity in bone cements, CS, and GO characteristics is given in Table 2. CS has a beneficial contribution to improving the biocompatibility and osseointegration of the ABCs, and it reduces the mechanical properties significantly. On the other hand, the GO provides mechanical reinforcement and does not present cytotoxicity in most reported investigations. The combination of both components, individually [101,109] or like a nanocomposite [102], has shown synergy in the biological and mechanical properties since there were increases in mechanical properties, up to 69%, and in viability, proliferation, and cellular adhesion of human osteoblasts and MG-63 cell culture in contact with these cements. Additionally, Wistar rats parietal bone implants for 3 months showed more excellent osseointegration in the cements that contained both loads.

## 7. Antibacterial Properties

Implanted bone cements carry a high risk of infection when introduced into the body due to the possibility of biofilm adsorption of microorganisms on an inert surface, which usually requires multiple surgeries for treatment [115,116].

Chronic infection of joint prostheses requires surgical removal of the implant to eradicate the infectious process. The procedure can be performed in a single or two steps. In the first place, the intervention consists of removing the infected implant, decontamination of the site, and implanting a new revision prosthesis covered with bone cement with antibiotics. In the second place, the revision prosthesis placement phase is delayed for a pair of months (2–5 months), during which a bone spacer cement impregnated with an antibiotic is placed in the site [117].

In recent years, the increase in resistance of microorganisms to antibiotics has led to severe health problems since most of the bacteria causing the infection are resistant to at least one of the antibiotics that are generally used to eradicate the infection; therefore, worldwide efforts are ongoing to study new antimicrobial agents that can effectively inhibit the growth of bacteria [59].

### 7.1. Formation of the Biofilm

All biofilms originate from the same sequence of events, as shown in Figure 6. In clinical uses of biomaterials, the biomaterial surface is first covered with a conditioning film composed of macromolecules adsorbed from the biological environment in which the biomaterial is placed (Figure 6a). In the case of orthopedic biomaterials, such as PMMA, which contact the bone and come into direct contact with the blood, they absorb several proteins from the plasma before the first microorganism appears [118].

Microorganisms arrive at this conditioning surface through various transport mechanisms, such as diffusion, convection, or sedimentation (Figure 6b). Orthopedic implants can be infected by direct contamination during surgery. The initial adhesion of microorganisms is reversible and depends on the microbial cell surface’s general physiochemical characteristics, the biomaterial surface, and the biological fluid. This reversible adhesion of microorganisms can be irreversible by producing an exopolymer (extracellular polymeric secretion produced by bacteria), which generates a firm anchorage of the bacteria (Figure 6c). After that, the exopolymers embedded microorganisms in the biofilm to form a called “glycocalyx” (Figure 6d). This term refers to the accumulation of glycoproteins in the external coating of the biomaterial, which, in addition to serving as an anchorage to biofilm, provides a physical and impenetrable barrier for antibiotics and immune cells [116,118].

The growth of the adhering organisms is the primary mechanism of multiplication in a biofilm to a newly added layer, and the bacteria find spaces to multiply, which finally leads to the formation of a denser film. As a final step in forming biofilms, organisms on the biofilm periphery can separate, leading to septic loosening [118].

### 7.2. Antimicrobial Properties in Acrylic Bone Cements

Open surgeries always risk contamination; however, biomaterials’ presence increases infection risk due to their susceptibility to bacterial colonization [118]. This event depends on several aspects, such as the physicochemical properties of the surface of the biomaterials, the structure of the cell membrane, and the bacteria receptors, which will define if the bacteria that reach the surface of the biomaterial can easily adhere and proliferate there, causing the septic loosening of the prosthesis [119].

Bacterial infections are a common complication after total joint replacement [120], with infection rates between 1% and 3% [117,118], mainly caused by *Staphylococcus epidermidis*, *Staphylococcus aureus*, and *Staphylococcus capitis* [97,121,122]. Their incidence is expected to increase significantly due to the expanded use of these procedures in older patients and younger people [123]. One way to reduce such infections is by adding antimicrobial agents to the bone cements used for implant fixation [120]. Commonly used antibiotics are tobramycin, gentamicin [122,124], vancomycin, and cephalosporins [125].

In antibiotic-loaded bone cements (ALBC), it is known that most of the antibiotic is retained inside the cement, and only a small amount on the surface is available for diffusion to the desired site [115]. The kinetics of antibiotic elution from bone cements consists of an initial phase of a rapid release of the drug, followed by a much slower release [126].

Some studies have emphasized that antibiotics are mainly released from the surface. The initial fast release is mainly due to the antibiotic dissolution from the surface or near the surface. Subsequently, some antibiotic is released slowly due to the antibiotic diffusion through the cracks, pores, and cement’s imperfections [118,127,128]. However, elution of the antibiotic incorporated into the cement is not complete since all in vitro studies indicate that only 5–8% of the total antibiotic incorporated into the ABCs is released after long periods, and in vivo studies also confirm that only 5–18% of the incorporated gentamicin is released [118]. Generally, the release kinetics correlated with the cement’s degree of porosity rather than the number of antibiotics [119].

ALBC has been widely used for the treatment of infected arthroplasty, and given the increased incidence of this type of infection, a large number of review papers is found in which aspects such as release mechanisms, clinical efficacy, and safety of the ALBC are discussed [13,118,129,130,131,132,133].

Bioabsorbable fillers such as poly(*N*-vinyl-2-pyrrolidone) [134], lactose, hydroxypropyl methylcellulose [128], gelatin microparticles [89], mesoporous silica nanoparticles [135], and carbon nanotubes [115] have been introduced in their formulations to improve antibiotic’s release from ABCs. Their dissolution and porous structure increase the open pores’ interconnection with the surface, and the antibiotic can be released into the surrounding material by molecular diffusion. This interconnection of the porosity increases the cement’s permeability and leads to an increase in the penetration of fluids, and thus, the antibiotics contained inside the ABCs will be available for elution [126]. These additions generated an increase in the proportion of released antibiotic; however, they produced a loss in the ABCs’ mechanical properties. This cement’s clinical use is restricted only to treating infections and not to the prostheses fixation [126].

The use of ALBC is a common practice in arthroplasty; however, the development of antibiotic-resistant bacteria due to the prolonged use of antibiotics makes the treatment of infections even more difficult [132,136]. Additionally, losses in mechanical properties due to the increase in the number of antibiotics that surgeons generally use to reduce the risk of infections rise, generating a premature implant with failures [26,137,138]. Therefore, it is necessary to develop alternatives to antibiotics for the treatment of infections.

The resistant bacterial species list includes multidrug resistant (MDR) and extremely drug-resistant (XDR) bacteria. These bacterial strains have evolved to exhibit resistance to almost all commercially available antibiotics, affecting millions worldwide [139]. Because of this, there are reports of the addition to ABCs with antimicrobial agents, such as quaternary amine dimethacrylate iodine comonomer [140], quaternary amine dimethacrylate comonomer [141], gold nanoparticles [142], 2-methacryloyloxyethyl phosphorylcholine, quaternary ammonium dimethylaminohexadecyl methacrylate [143], and Ag_2_O [144], different from the antibiotics.

### 7.3. Antibacterial Mechanism of Chitosan

The antimicrobial activity of CS depends strongly on physical properties such as M_w_ and DD. CS with higher DD tends to have higher antimicrobial activity [145,146], while with M_w_, it has the opposite effect [147]. Although some authors report that maintaining constant DD, there is an optimal value of M_w_ in which the highest antimicrobial activity is obtained (90,000 Da) [148]. Others reported that an M_w_ range between 10,000 and 100,000 Da is adequate to inhibit *Escherichia coli*, while a molecular weight of 2200 Da accelerated bacterial growth. Concerning CS concentration, it is found that low concentrations (0.2 mg/mL) favored the bonding of polycation to the cell’s negative surface, causing its impermeability, while high concentrations could generate a positively charged network by remaining in suspension [149].

The exact mechanism of the antimicrobial activity of CS is not well known. However, so far, the most accepted is based on the electrostatic interactions between positively charged chitosan (NH_3_^+^) and the negatively charged components of the bacterial cell membrane, such as the anionic cell wall of glycans and proteins or phospholipids in the cytoplasmic membrane. Depending on the type and location of the interaction, various effects are possible. The adhesion of chitosan to the anionic macromolecules of the cell wall forms an impermeable layer around the cell, preventing the transport of nutrients into and out of the cell. In contrast, the interaction with cell membrane constituents alters the permeability, leading to the leakage of intracellular electrolytes, glucose, enzymes, and other low-molecular-weight proteinaceous cytoplasmic materials [149,150,151].

The CS interrupts the physiological activities of the cell but affects *Gram-positive* and *Gram-negative* cells differently since the CS with higher molecular weight forms a polymer shield to prevent the entry and exit of nutrients in *Gram-positive* cells, and with low M_w_ enters the *Gram-negative* cell, and binds to cytoplasmic compounds, causing flocculation with altered physiological processes [151].

Other possible antibacterial mechanisms of CS include chelation of essential nutrients necessary for cell growth or inhibition of messenger RNA and protein synthesis through penetration of chitosan into the cell nucleus of the microorganism and the bonding of CS to DNA. For this penetration to occur, CS must have an M_w_ below 5000 Da [147], which allows it to pass through the bacterial cell wall, composed of multilayers of cross-linked murein in the plasma membrane [150,151]. The minimum concentration of CS inhibition is between 0.05% and 0.1%, depending on the molecular weight of bacteria and the CS [152].

#### Antibacterial Effect of Chitosan in Acrylic Bone Cements

The addition of CS to ABCs has shown a percentage-dependent antibacterial effect. Tunney et al. [96] and Dunne et al. [97] added between 1 and 5 wt.% of chitosan in the solid phase of the cement and found no antibacterial activity against *S. epidermidis*, *S. aureus*, and *S. capitis*. However, Valencia et al. [101,109] and Shi et al. [120] report antibacterial activity against *S. aureus*, *S. Epidermidis,* and *E. coli* of cements added with 15% d of CS.

Researches indicate that using chitosan nanoparticles and quaternary ammonium nanoparticles derived from CS as bactericides in the commercial cements CMW Smartset and CMW Smartset-G against *S. aureus* and *S. epidermidis* have found that these nanoparticles generate inhibition of the growth of both bacteria. Besides, they do not present toxic effects on 3T3 mouse fibroblasts, and the mechanical properties of both cements are maintained [120]. Wang et al. [46] found that the addition of quaternary chitosan (*N*-(2-hydroxy)propyl-3-trimethylammonium chitosan chloride)-based hydrogels loaded with nanosized hydroxyapatite gave the cement excellent antibacterial properties against *E. coli* and *S. aureus* bacteria.

Other studies show the effect of including quaternary chitosan as hydroxypropyltrimethyl ammonium chloride (HACC) in the ABCs, which inhibits the development of bone infections caused by antibiotic-resistant bacteria [136]. The effects also include lowering T_max_, prolonging t_set_, increasing hydrophilicity, increasing apatite formation when immersed in SBF, and improving adhesion and proliferation of human bone marrow-derived mesenchymal stem cells [153].

In the last years, there has been a great interest in reinforcing polymeric matrices with nanomaterials to generate biocompatible and antibacterial nanocomposites [154]. In the case of CS, it is found that it has an excellent antimicrobial activity, and it is improved against a wide range of *Gram-positive* and *Gram-negative* bacteria with the incorporation of nanoparticles, such as zinc oxide (ZnO) [155,156], Au [4,157], Ag [45,158,159], and reduced GO [94]. Some of these nanocomposites in ABCs demonstrated improvement in antibacterial activity compared to CS alone [4,45]. Future studies could include incorporating CS nanocomposites that lack evaluation in ABCs to study their effect on the cement’s antibacterial properties.

### 7.4. Antibacterial Mechanism of Graphene Oxide

GO is a promising material for developing antimicrobial and antiviral surfaces due to its high surface/volume ratio [160]. The antimicrobial activity of GO is contact based; thus, it offers a viable alternative to surfaces that release biocides such as antibiotics, which are consumed at the surface over time [161]. It has been proposed that the antimicrobial activity of GO occurs when films come into direct contact with bacterial cells through physical and chemical interactions due to their negative charge and laminar structure [49,62,160,162]. In this process, the cell membrane is the main target of GO cytotoxicity since membrane damage has been evidenced in bacteria exposed to GO through morphological changes in cell structure, leakage of ribonucleic acid (RNA), intracellular electrolytes, absorption of dyes impermeable to the membrane, and changes in transmembrane potential [161].

Membrane damage can be caused by the sharp atomic edges of GO, which can penetrate the cell membrane and physically disrupt its integrity, or it can also be produced through lipid peroxidation induced by the oxidizing nature of GO. Oxidative stress has been considered an essential component of antimicrobial activity for bacterial cells exposed to GO [49,161]. In Table 3, the different mechanisms of antimicrobial activity proposed for GO are highlighted.

Several researchers suggest that the sharp edges of the GO and the reactive oxygen species (ROS) are not a fundamental part of the antimicrobial mechanism of GO. Instead, the primary mechanism considered is transferring electrons from the bacterial membrane to the GO’s surface. When it comes to bacteria, GO acts as an electron receptor that pumps the electron outside the bacteria’s membrane, creating oxidative stress independent of the ROS. The antimicrobial activity of GO is strongly related to the number of basal planes of the sheets [165].

The antimicrobial activity of GO and other allotropes of carbon is affected by the nanomaterial’s surface’s size and chemistry. An increase in the available surface area through particle size reduction improves antibacterial activity due to more significant interaction with bacteria [59]. This behavior was reported by Perreault et al. [161], who worked with GO films on which it was proved that the antimicrobial activity against *E. coli* bacteria is inversely related to the size of the sheets, reaching an increase in this activity of four times when the area of the sheet varied from 0.65 to 0.01 m^2^. This higher antimicrobial activity of the smaller GO sheets is attributed to the oxidative mechanism associated with the smaller sheets’ higher defect density. In contrast, GO suspensions results show that antimicrobial activity increases with sheet size [106,161] against the same bacteria.

The antimicrobial activity of GO has also been questioned by researchers who affirmed not to have found any type of intrinsic inhibition in the growth of *E. coli* with GO [167], nor *Pseudomonas aeruginosa* [94]. However, they suggest that GO is a cell growth stimulator by increasing the adhesion and proliferation of mammalian cells [167]. These contradictory results encourage further research on the effects of GO on microorganisms. However, because materials in the graphene family have generally demonstrated bacterial toxicity and relatively low cytotoxicity, it has been suggested that these materials can be applied as antimicrobial products [168].

#### Antibacterial Effect of GO on ABCs

When evaluating the antibacterial activity of cements added with GO against *Gram-positive* bacteria *Bacillus cereus* and *S. aureus* and *Gram-negative Salmonella enterica* and *E. coli*, it was found that the increase in GO content up to 0.5 wt.% increased this property significantly in the cement with both types of bacteria with a level of significantly lower than 0.01 [108]. Likewise, in other investigations, it has been confirmed the antibacterial activity of bone cements against *E. coli*, with GO percentages of 0.3 wt.% [101,109]. In these investigations, it was reported that although the significant contribution to the antibacterial activity of ABCs loaded with GO and CS is generated by GO, CS also has a minor contribution to this property.

On the other hand, according to the results presented by Paz et al. [114], ABCs modified with 0.1 wt.% of GO showed a lack of antimicrobial activity against *S. Aureus*, possibly because the GO powder level was below the threshold required for a positive antimicrobial response. This confirms the results obtained by several authors, who affirmed that GO’s antibacterial activity depends on the concentration [108,163,164].

Table 4 shows a summary of the strategies used to confer antimicrobial activity to bone cements and the characteristics offered by CS and GO. From this table, we conclude that the development of bacteria resistant to antibiotics due to their excessive use has generated massive research potential towards using new agents that generate antimicrobial activity in ABCs. Despite the good antibacterial properties conferred to CS nanometric size cements or modified with some functionalization, GO has conferred to cement outstanding antimicrobial properties. This makes that cements added with GO or GO + CS are promising for preventing and treating infections in arthroplasties.

## 8. Future Perspectives

Taking into consideration the excellent bioactive properties shown by the ABC added with CS and GO, future research can focus on the factors affecting the residual monomer content in these formulations in order to reduce this property and the risk of generating chemical necrosis of the surrounding tissue and thus loosening of the prosthesis.

The good antibacterial properties of cements loaded with GO and CS against *Gram-positive* and *Gram-negative* bacteria studied so far are promising results. An in-depth study of the potential of this type of addition in developing bone cements with antibacterial properties is needed, broadening the range of bacteria studied and including those that occur more frequently after a total joint replacement.

The antibacterial properties of cements loaded with CS and GO are promising for their use in the prevention and treatment of infections generated in arthroplasties, so it would be of great interest to compare the effect and time of action loads concerning antibiotics commonly used in the treatment of these infections.

## 9. Conclusions

Despite ABCs being inert materials that do not stimulate any type of chemical adhesion with bone, the incorporation of CS has improved the biological interaction in cements, increasing cell viability, growth, and cell adhesion in osteoblasts, and increasing the deposition of Ca and P ions on the surface of the cement under in vitro conditions. It also prevents fibrous tissue formation around the cement and promotes osseointegration with bone under in vivo conditions in animal biomodels. The contribution of CS to the antibacterial properties of cements has been limited and increased when its size is reduced to the nanometric scale or when it has determined functionalities. However, the mechanical properties of these cements are decreased.

ABCs loaded with GO show high potential for use in preventing and treating infections in arthroplasties. They have shown elevated antibacterial activity against some types of *Gram-positive* and *Gram-negative* bacteria and have also been shown to promote osteogenesis.

The addition to the ABCs of CS and GO could be an attractive alternative to generate bioactive and antibacterial ABCs with adequate mechanical properties since GO has contributed to improving the mechanical properties with a reinforcing effect in the cements loaded with CS.

## Figures and Tables

**Figure 1 biomolecules-10-01616-f001:**
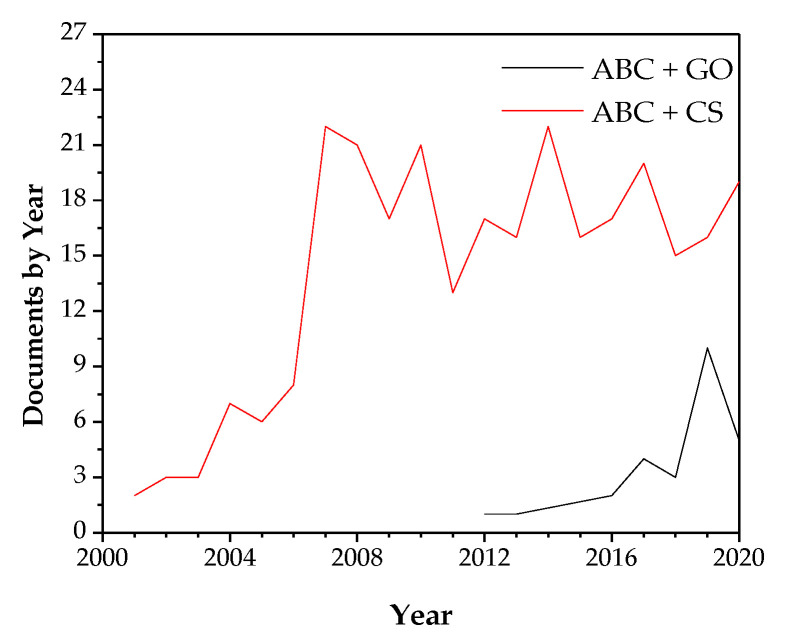
The number of documents published in 2001–2020 about acrylic bone cements loaded with chitosan and graphene oxide. The keywords used were (“Bone cement” AND “Graphene oxide”) and (“Bone cement” AND chitosan)—source: Scopus (19 October 2020).

**Figure 2 biomolecules-10-01616-f002:**
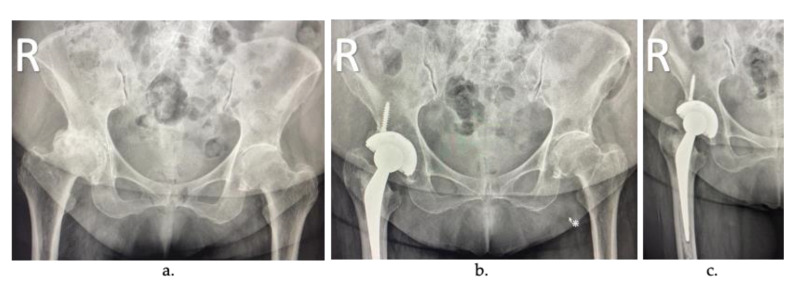
Arthroplasty radiographs: (**a**) patient with severe osteoarthritis in the right hip, (**b**) joint replacement of the femoral component, and (**c**) cemented joint replacement.

**Figure 3 biomolecules-10-01616-f003:**
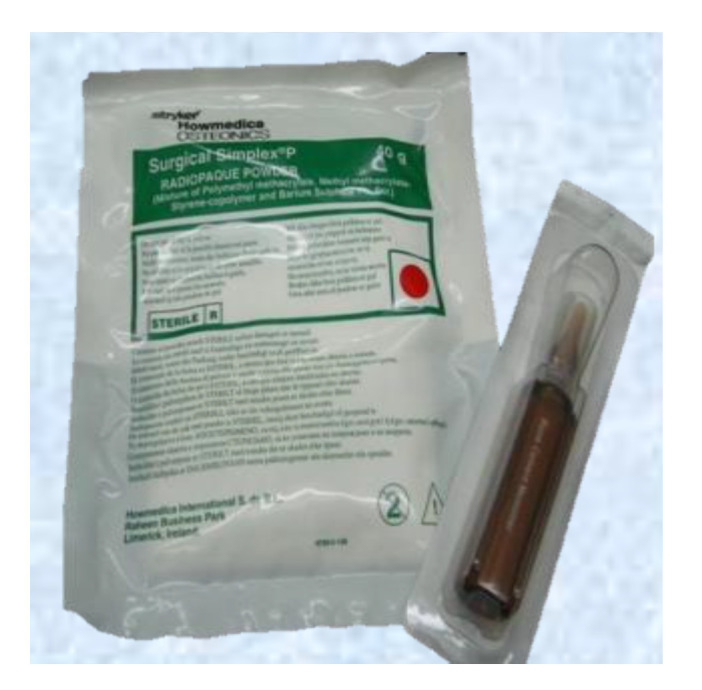
Components of commercial acrylic bone cement (ABC) Surgical Simplex P. Solid phase in a sterile plastic package and liquid phase in a sterile glass vial.

**Figure 4 biomolecules-10-01616-f004:**
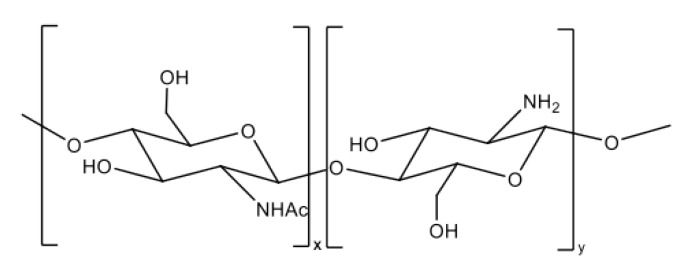
Chemical structure of chitosan.

**Figure 5 biomolecules-10-01616-f005:**
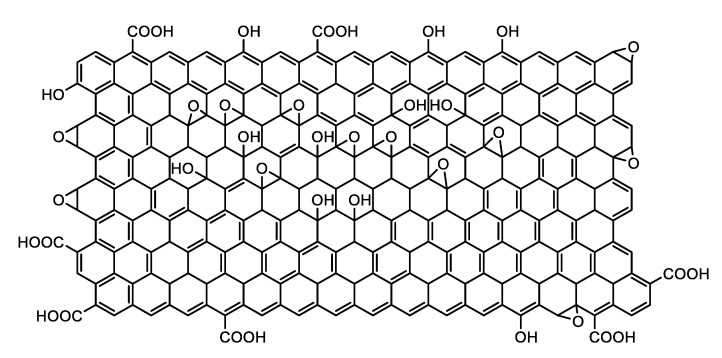
Chemical structure of graphene oxide (GO).

**Figure 6 biomolecules-10-01616-f006:**
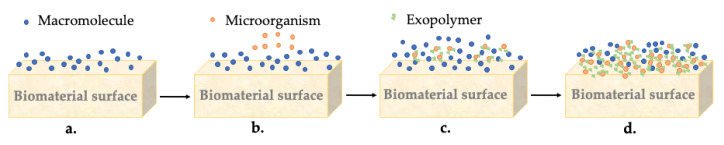
Description of the biofilm formation: (**a**) formation of the conditioning film, (**b**) initial adhesion of the first microorganisms, (**c**) production of exopolymer, and (**d**) glycocalyx formation.

**Table 1 biomolecules-10-01616-t001:** Essential components of acrylic bone cement formulations.

Component	Composition (~wt.%)	Function	Structural Formula
Solid Phase			
PMMA	87.5–89.25	Polymer	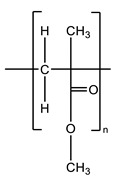
Barium sulfate or zirconium dioxide	10	Radiopaque agent	BaSO_4_ or ZrO_2_
Benzoyl peroxide	0.75–2.5	Polymerization reaction initiator	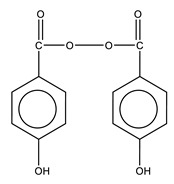
Liquid Phase			
MMA	97.0–97.5	Monomer	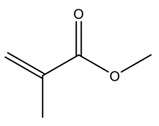
*N,N*-Dimethyl-*p*-toluidine	2.0–2.5	Room-temperature polymerization reaction accelerator	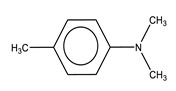
Hydroquinone	75 ppm	Inhibitor that prevents premature polymerization of MMA	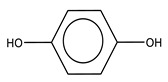

**Table 2 biomolecules-10-01616-t002:** Summary of the effect of chitosan (CS) and graphene oxide (GO) on the bioactivity of Acrylic Bone Cements (ABCs).

Materials	Advantages	Disadvantages	References
ABC	Inert behavior	Insufficient adhesion to the bone	[10,14,32]
ABC + chitosan	Porous ABCs Improvement in osseointegration Reduction in T_max_ reached during polymerization Increasing cellular activity of osteoblasts	Reduction in mechanical properties Increasing residual monomer content	[4,16,45,92,95,96,97,98,99,100,101,102,103]
ABC + graphene oxide	Improvement of mechanical properties Reduction T_max_ reached during polymerization Good compatibility with bone tissue Accelerates bone formation Noncytotoxic to human osteoblasts, L929 fibroblasts, MG-63, hBMSCs, and MC3-T3 cells.	Increasing residual monomer content Reduction in cell viability against MG-63 cell culture.	[22,101,102,107,108,109,110,111,112,113,114]
ABC + chitosan + graphene oxide	Increased osseointegration Increased mechanical properties Increased viability and cell adhesion (MG-63 and human osteoblast) Increased antibacterial activity compared to separate loads	Increase in residual monomer content	[101,102,109]

**Table 3 biomolecules-10-01616-t003:** Primary mechanisms of antimicrobial activity in GO.

Proposed Mechanism	Definition	References
Extremely sharp edges	It is the primary mechanism of antimicrobial activity in GO. Damage to the membrane when it comes into contact with the edges.	[59,163]
Oxidative stress (through the production of ROS)	It plays a minor role in antimicrobial activity. GO induces reactive oxygen species (ROS), which disrupts the balance in redox processes inside the cell and causes damage to cell components leading to apoptosis in bacteria.	[164]
Oxidative stress (independent of ROS production)	GO sheets interrupt a specific microbial process by disturbing or oxidizing a vital cell component or structure without producing ROS.	[50,165]
Formation of GO-cell aggregates	GO sheets envelop the bacteria forming aggregates, locally disturbing the cell membrane, and inducing the decrease in the bacterial membrane potential and the leakage of fungal spore electrolytes.	[59,166]

**Table 4 biomolecules-10-01616-t004:** Summary of strategies employed to confer antimicrobial activity in ABCs and the effect of CS and GO on these cement properties.

Strategies	Studied Bacteria	Advantages	Disadvantages	References
ABC + antibiotics	*S. Aureus*	Releases the antibiotic	Most of the antibiotic is trapped inside the cement Antibiotic-resistant bacteria have been developed	[118,124,125,126,127]
ABC + antibiotics + bioactive fillers	---	A higher percentage of antibiotic is released Improves osseointegration	Reduces mechanical properties Antibiotic-resistant bacteria have been developed	[89,115,126,128,134,136]
ABC + chitosan	*S. epidermidis* *S. Aureus* *E. coli* *S. Capitis*	Presents good antimicrobial activity when the CS is functionalized or in nanosize.	CS without modifications Presents extremely low or no antimicrobial activity in ABCs	[96,97,101,109,120,136,153]
ABC + graphene oxide	*B. cereus* *S. Aureus* *S. enterica* *E. coli*	Shows antimicrobial activity against *Gram-positive* and *Gram-negative* bacteria		[101,108,109,114]
ABC + chitosan + graphene oxide	*E. coli*	Increased antibacterial activity compared to cement with separately added		[101,109]

--- Paper does not report studies with bacteria; it only reports the antibiotic elution improvement.

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
