# Peer review of "The Role of Chitosan and Graphene Oxide in Bioactive and Antibacterial Properties of Acrylic Bone Cements"

_biomolecules, 2020, doi:10.3390/biom10121616_

Round 1

Reviewer 1 Report

The subject of the Review covers benefits of acrylic bone cements with chitosan and graphene oxide addition which resulting in formulations with great potential for use in orthopedics. The authors conclude that combination of these additives could be an attractive to generate bioactive and antibacterial ABCs with adequate mechanical properties. The authors also competently highlighted future perspectives associated with an in-depth study of the different related factors to decrease the disadvantages of such composites. The Manuscript is logically structured and will be of interest to a wide range of the Journal’s readers. The text should be revised a little. The comments are shown as follows:

1) Line 96: Please uncover BPO and DMPT. “The polymerization process begins with BPO and DMPT mixture with formation of benzoyl radicals”? Benzoyl peroxide itself serves as a source of benzoyl radicals.

2) The quality of the structures in Table 1 should be improved.

3) Line 148: “O-acetylation” instead “o-acetylation”

4) Line 198: “…of the cement” instead “…and the cement”

5) Line 324: Table 2 should be moved to line 310 (before Antibacterial properties unit)

Author Response

Reviewer 1

Comments and Suggestions for Authors

The subject of the Review covers benefits of acrylic bone cements with chitosan and graphene oxide addition which resulting in formulations with great potential for use in orthopedics. The authors conclude that combination of these additives could be an attractive to generate bioactive and antibacterial ABCs with adequate mechanical properties. The authors also competently highlighted future perspectives associated with an in-depth study of the different related factors to decrease the disadvantages of such composites. The manuscript is logically structured and will be of interest to a wide range of the Journal's readers. The text should be revised a little. The comments are shown as follows:

  1. Line 96: Please uncover BPO and DMPT. "The polymerization process begins with BPO and DMPT mixture withformation of benzoyl radicals"? Benzoyl peroxide itself serves as a source of benzoyl radicals.

R// We appreciate the reviewer's comment. BPO and DMPT were uncovered, and the text was modified. "The polymerization process begins after mixing of the benzoyl peroxide (BPO) (in the solid) and N,N-Dimethyl-p-Toluidine (DMPT) (in the liquid) with the production of benzoyl radicals at room temperature [18,19]"

  1. The quality of the structures in Table 1 should be improved.

R// We appreciate the reviewer's comment. Structures in table 1 were improved.

  1. Line 148: “O-acetylation” instead “o-acetylation”

R// We appreciate the reviewer's comment. Letter o was changed.

  1. Line 198: "…of the cement" instead "…and the cement"

R// We appreciate the reviewer's comment. The sentence was changed to: “With this type of inorganic filler, it is necessary to control the amount used since they affect the cement's mechanical properties [32,75].”

  1. Line 324: Table 2 should be moved to line 310 (before Antibacterial properties unit)

R// We appreciate the reviewer's comment. Table 2 was moved to line 314.

Reviewer 2 Report

In ‘The Role of Chitosan and Graphene Oxide in Bioactive and Antibacterial Properties of Acrylic Bone Cements’, Valencia Zapata et al. review this topic with an emphasis on the improvement of biological properties.

Major comment:

In section 7.3.1, sometimes the addition of chitosan seems to have an antibacterial effect and sometimes it doesn’t. It is not completely clear from the review how these studies are different or why these differences have been observed. Some more details about the specific materials being tested may be helpful here.

Minor comments:

Check that all abbreviations are defined in the main text.

Check Table 1 that some of the structures do not overlap.

Line 199: module à modulus

Line 277: Do you mean chemical necrosis instead of thermal necrosis?

In particular, section 6.3 should be revised for grammar.

Line 285-286: What were the anabolic and catabolic effects?

Lines 296-299: This is an unclear sentence.

Line 379: What is meant by bioabsorbable charges?

Line 445: What are rabbit infections tibia?

Author Response

Reviewer 2

Comments and Suggestions for Authors

In 'The Role of Chitosan and Graphene Oxide in Bioactive and Antibacterial Properties of Acrylic Bone Cements', Valencia Zapata et al. review this topic with an emphasis on the improvement of biological properties.

Major comment:

In section 7.3.1, sometimes the addition of chitosan seems to have an antibacterial effect and sometimes it doesn't. It is not completely clear from the review how these studies are different or why these differences have been observed. Some more details about the specific materials being tested may be helpful here.

R// We appreciate the reviewer's comment. Paragraph 1 of section 7.3.1 was rewritten as follows "The addition of CS to ABCs has shown a percentage-dependent antibacterial effect. Tunney et al. [96] and Dunne et al. [97] added between 1 and 5 wt.% of chitosan in the solid phase of the cement and found no antibacterial activity against S. epidermidis S. aureus, and S. capitis. However, Valencia et al. [101,109] and Shi et al. [120] report antibacterial activity against S. aureus, S. Epidermidis, and E. Coli of cements added with 15% d of CS."

Minor comments:

  1. Check that all abbreviations are defined in the main text.

R// We appreciate the reviewer's comment. All abbreviations were revised and defined in the main text.

  1. Check Table 1 that some of the structures do not overlap.

R// We appreciate the reviewer's comment. Structures in table 1 were corrected.

  1. Line 199: module à modulus

R// We appreciate the reviewer's comment. The word module was changed by modulus.

  1. Line 277: Do you mean chemical necrosis instead of thermal necrosis?

R// We appreciate the reviewer's comment. The word thermal was changed by "chemical."

  1. In particular, section 6.3 should be revised for grammar.

R// We appreciate the reviewer's comment. Grammar in section 6.3 was corrected.

  1. Line 285-286: What were the anabolic and catabolic effects?

R// We appreciate the reviewer's comment. The sentence was rewritten as "Mirza et al. [113] prepared ABCs with GO and evaluated human bone marrow mesenchymal stem cells (hBMSCs). The anabolic genes (COL1A1, BMP4, BMP2, RUNX2, and ALP) were associated with stimulatory effects, while the catabolic genes (MMP2 and MMP9) exhibit inhibitory in the presence of GO."

  1. Lines 296-299: This is an unclear sentence.

R// We appreciate the reviewer's comment. These lines were rewritten as follows "Contrary to all the results shown above, Sharma et al. [22] compared cements' osseointegration with GO, graphene (G), and amine-functionalized Graphene (AG), showing a reduction of cell viability against human osteoblast (MG-63) in the ABCs loaded with GO and G compared to the cement containing AG. Then, cell apoptosis was generated because GO and G penetrated the cytoplasm by endocytosis pathway, disturbing the metabolic activity, gene transcription, and translation processes."

  1. Line 379: What is meant by bioabsorbable charges?

R// We appreciate the reviewer's comment. The word "charges" was changed by fillers.

  1. Line 445: What are rabbit infections tibia?

R// We appreciate the reviewer's comment. That sentence was changed by bone infections.

Reviewer 3 Report

Authors should improved the chapter on future perspective.

Author Response

Reviewer 3

Comments and Suggestions for Authors

  1. Authors should improve the chapter on future perspective.

R// We appreciate the reviewer's comment. The chapter on future perspective was improved
